# Metabolomic Analysis of Germinated Brown Rice at Different Germination Stages

**DOI:** 10.3390/foods9081130

**Published:** 2020-08-17

**Authors:** Hoon Kim, Oui-Woung Kim, Jae-Hwan Ahn, Bo-Min Kim, Juhong Oh, Hyun-Jin Kim

**Affiliations:** 1Korea Food Research Institute, Research Group of Consumer Safety, 245 Nongsaengmyeong-ro, Iseo-myeon, Wanju-Gun, Jeollabuk-do 55365, Korea; hkim@kfri.re.kr (H.K.); kwoung@kfri.re.kr (O.-W.K.); jhahn@kfri.re.kr (J.-H.A.); 2EZmass.Co. Ltd., 501 Jinjudaero, Jinju, Gyeongsangnam-do 52828, Korea; qhals159@naver.com (B.-M.K.); j_h_5@naver.com (J.O.); 3Division of Applied Life Sciences (BK21 plus), Department of Food Science and Technology, and Institute of Agriculture and Life Science, Gyeongsang National University, 501 Jinjudaero, Jinju, Gyeongsangnam-do 52828, Korea

**Keywords:** brown rice, germination, metabolomics, GC-MS, LC-MS

## Abstract

Brown rice (BR) is unpolished rice containing many bioactive compounds in addition to the basic nutrients of the rice grain. Herein, BR was germinated for up to 48 h to prepare germinated brown rice (GBR). The physiological and chemical changes in the GBR during germination were analyzed. GBR samples germinated for 48 h were in the radicle-emergence stage, but root formation was not observed. The change in the GBR metabolite profile during germination was analyzed to determine the effect of germination on the chemical profiles of the GBR samples. Twenty-five metabolites including acidic compounds, amino acids, sugars, lipid metabolites, and secondary metabolites were identified as the components that contributed to the variations in the GBR groups germinated for different time periods. Among the metabolites, the carbohydrates associated with energy production and lipid metabolites changed significantly. Based on the identified metabolites, a metabolomic pathway was proposed. Carbohydrate metabolism, citric acid cycle, and lipid metabolism were the main processes that were affected during germination. Although further studies on the relationship between the metabolite profile and nutritional quality of the GBR are needed, these results are useful for understanding the effect of germination on the physiological and chemical changes in BR.

## 1. Introduction

Rice (*Oryza sativa* L.), one of the most important cereal grains, is a staple food for over 50% of the global population [1]. Although rice is a good source of protein, amino acids, fiber, iron, vitamins, and minerals, many nutrients are removed during the milling of rice because most nutrients are present in the outer bran layer of rice [2,3]. Unlike white rice without the outer bran layer, brown rice (BR), an unpolished whole-grain rice, contains not only the basic nutrients of the rice grain but also many bioactive compounds including ferulic acid and gamma aminobutyric acid (GABA) [4]. However, BR is not considered a suitable staple food owing to its rough texture and difficulty in cooking [5].

Germinated brown rice (GBR) was developed to overcome these problems. To prepare GBR, BR is usually soaked for about 10–12 h in warm water and after rinsing, it is kept moist for 20–24 h, and water is changed every 3–4 h [6]. This simple and inexpensive process not only softens the texture of BR but also enriches the nutrients. In particular, many animal and clinical studies suggested that the intake of GBR prevented against hyperlipidemia, hypertension, and hyperglycemia, and showed a positive correlation to the reduction in the risk of some chronic diseases such as cancer, diabetes, cardiovascular disease, and Alzheimer’s disease [4,6]. Moreover, GBR flour was found to improve bread quality when substituted for wheat flour [7].

Recently, to better understand the relationship between germination and the nutritional quality of grains including GBR, various omics technologies such as genomics, proteomics, and metabolomics have been applied [8,9,10,11,12]. Among these technologies, metabolomics is a useful tool to determine the chemical change during germination. Based on the seed germination or cultivation methods, the metabolomic profiles of the grains including rice [13,14], wheat [15,16,17], and soybean [18] were analyzed using liquid chromatography–mass spectrometry (LC-MS), gas chromatography–mass spectrometry (GC-MS), and/or nuclear magnetic resonance (NMR), which afforded useful information regarding the changes in nutritional components during germination. However, a metabolomic analysis of the GBR has rarely been performed with the exception of the GC-MS-based metabolomic study on three different BR kernels [19]. During germination for 96 h, 174 metabolites in the BR kernels changed significantly. This study provided information on the correlation between germination and nutritional quality of BR kernels. However, the edible GBR typically germinates within 24 h or 48 h, but not 96 h. In general, GBR germinated for more than 48 h develops roots as well as shoots, and there is a possibility that significantly more chemicals are modified than expected. 

Therefore, in this study, BR was germinated for 48 h and the metabolite profiles of the GBR samples were collected every 4 h and analyzed using LC-MS and GC-MS to obtain detailed information regarding the changes in the metabolites. Differences in the samples were visualized by the partial least-squares discriminant analysis (PLS-DA) model. The major metabolites contributing to these differences were identified, and a metabolomic pathway was proposed. 

## 2. Materials and Methods

### 2.1. BR and Germination

The rice used in this study was of the Chucheong variety, with short grains, and was harvested in Hwaseong, Korea in 2017. After the harvest, the inedible outer hull of the rice was removed, the BR was dried using a dryer (INBD-150E, Hansung, Iksan-shi, Korea) at 35 °C for 17 h, and the moisture content was determined as 15.9%. To prepare GBR, the dried BR were soaked in water in the fermenter (180 L, Rubbermaid Korea, Seoul, Korea) at 20 °C and then fine air bubbles were continuously provided by the air pump of the fermenter for 48 h. During germination, GBR samples were collected after 4, 8, 12, 16, 24, 32, 40, and 48 h, and non-germinated BR was used as a control (0 h). 

### 2.2. Determination of Moisture Content, Germination Rate, and Seedling Length

The moisture content of 200 kernels during germination was determined using a single-kernel moisture meter (PQ-520, Kett, Tokyo, Japan). The germination rate was determined by visually counting the number of germinated seeds in 100 kernels. The seeding length of the germinated BR was measured using a scale lupe (7X, Peak, Tokyo, Japan) for 30 kernels. All measurements were conducted in triplicate.

### 2.3. Sample Preparation for Metabolomic Analysis

Each lyophilized GBR sample was homogenized with 50% methanol using a bullet blender (Next Advance, Troy, NY, USA). After centrifugation, the supernatant was completely dried by a CentriVap concentrator SpeedVac (Labconco Co., Kansas City, MO, USA). For LC-MS analysis, the residues were separated with 20% methanol using terfenadine as an internal standard. After centrifuging, the supernatants were analyzed by ultra-performance liquid chromatography quadrupole time of flight MS (UPLC-Q-TOF MS, Waters Corp., Milford, MA, USA). For GC-MS analysis, all dried samples were dissolved in 70 μL of methoxyamine hydrochloride in pyridine (20 mg/mL) containing dicyclohexyl phthalate as an internal standard and incubated at 37 °C for 90 min. The methoxylated samples were then derivatized using 70 μL N,O-bis(trimethylsilyl)trifluoroacetamide with 1% trimethylchlorosilane at 70 °C for 30 min. The derivatized samples were analyzed by GC-MS (Shimadzu Corp., Kyoto, Japan).

### 2.4. Analysis of GBR Metabolites Using UPLC-Q-TOF MS 

GBR metabolites were analyzed using a UPLC-Q-TOF MS system (Xevo G2-S; Waters, Milford, MA, USA) [18]. The samples were injected into an Acquity UPLC BEH C_18_ column (2.1 mm × 100 mm, 1.7 μm; Waters Corp., Milford, MA, USA), equilibrated with water containing 0.1% formic acid, and eluted with a gradient of acetonitrile containing 0.1% formic acid. A flow rate of 0.35 mL/min and column temperature of 40 °C were employed. The eluted metabolites were analyzed using the Q-TOF MS system with positive electrospray ionization (ESI) mode. The desolvation flow rate and temperature were 800 L/h and 400 °C, respectively, and the source temperature was 100 °C. A TOF MS scan range of 100–1500 m/z, scan time of 0.2 s, and capillary and sampling cone voltages of 3 kV and 40 V, respectively, were used. Leucine-enkephalin ([M + H] = 556.2771), used as a lock mass, was infused at a flow rate of 20 μL/min and frequency of 10 s to ensure mass measurement accuracy of the metabolites analyzed by the instrument. A quality-control (QC) sample prepared by mixing all samples was analyzed in triplicate before the start and once after every 10 analyses. The MS/MS spectra were obtained in the range of m/z 50–1500 using a collision energy ramp from 10 to 30 eV. 

### 2.5. Analysis of GBR Metabolites Using GC-MS 

The GBR extracts were analyzed using a GC-2010 plus system (Shimadzu Corp., Kyoto, Japan) equipped with a DB-5ms capillary column (30 m × 0.25 mm, 0.25 µm, Agilent J&W column, Agilent Technologies, Santa Clara, CA, USA). The derivatized samples were injected into the column with split ratios of 1:50. Helium was used a carrier gas with a flow rate of 1 mL/min. The injector temperature was 200 °C. The oven temperature was maintained at 70 °C for 2 min, then successively increased at a rate of 5 °C/min to 150 °C, 3 °C/min to 210 °C, and 8 °C/min to 320 °C, which was finally held for 8 min. The eluents were detected using a GCMS-TQ 8030 MS (Shimadzu) system with electron ionization at 70 eV, ion source temperature of 230 °C, and interface temperature of 280 °C. The data were monitored and collected in the full-scan mode in the mass range of m/z 45–550 with a scan event time of 0.3 s and a scan rate of 2000 amu/s. The QC sample was analyzed once for every 10 samples. 

### 2.6. Data Processing 

The collection, normalization, and alignment of the MS dataset analyzed using the UPLC-Q-TOF MS were carried out using UNIFI version 1.8.2 (Waters Corp.). The peaks were collected using the peak-to-peak baseline noise as 1, a peak width at a 5% height of 1 s, noise elimination of 6, and an intensity threshold of 10,000. The collected data were aligned using a mass window of 0.05 Da and a retention time window of 0.2 min. Analyzer Pro application (Spectralworks Ltd., Runcorn, UK) was used to analyze the GC-MS data. The peaks were collected using an area threshold of 10,000, height threshold of 1, signal to noise ratio of 10, width threshold of 0.01, scan windows of 5, and smoothing of 5. The collected data were aligned with a retention time window of 0.1 min. All LC-MS and GC-MS data were normalized using the average mass intensity of the internal standards. The metabolites were identified based on the online databases (NIST 11 and Wiley 9 mass spectral libraries for GC-MS; ChemSpider database in UNIFI, METLIN database (www.metlin.scripps.edu), and human metabolome databases (www.hmdb.ca) for LC-MS), authentic standards, and retention indices (RIs) calculated using n-alkanes for GC-MS.

### 2.7. Statistical Analysis

SIMCA-P^+^ version 12.0.1 (Umetrics, Umeå, Sweden) was used to analyze the LC-MS and GC-MS data, and PLS-DA was used to visualize the differences in the sample groups. The quality of the PLS-DA model was evaluated using three parameters (R_2_X and R_2_Y: goodness of fit measure, Q_2_: predictive ability, and *p*-value) and was validated by the permutation test (*n* = 200). All data, including the normalized intensities of the identified metabolites, moisture content, germination ratio, and seedling length were statistically analyzed by one-way analysis of variance (ANOVA) with Duncan’s test (*p* < 0.05) using SPSS 17.0 (SPSS Inc., Chicago, IL, USA). 

## 3. Results and Discussion 

### 3.1. Germination Ratio, Seedling Length, and Moisture Content of GBR

The morphologies of the GBR samples are shown in Figure 1 and their general characteristics in terms of germination ratio, seedling length, and moisture content are shown in Figure 2. Seed germination begins with imbibition and ends with the emergence of the radicle from the seed coat [20]. In this study, the seedling known as coleoptile, mainly emerged from the BR embryo, which was swollen and brightened, at 8 h, whereas the radicle emerged at 48 h Figure 1. However, root formation was not observed at 48 h, unlike wheat [21] and barley [22] seed germinations. BR absorbed water up to 8 h (imbibition stage), whereas the BR samples germinated for 8–40 h were in the coleoptile-emergence stage and those germinated for 48 h were in the radicle-emergence stage. The BR that is germinated further, i.e., undergoes post-germination, is not suitable for eating. Germination ratio, seedling length, and moisture content increased with an increase in the germination period Figure 2. Germination ratio of 21% at 4 h increased by approximately 97% at 32 h and no additional increase was observed thereafter, whereas the seedling length increased continuously and the length at 48 h was approximately 2.3 mm. The moisture content increased by approximately two times at 4 h compared to the control (0 h, 15.9%), and gradually increased by 36.8% at 48 h. 

BR samples with 15.9% of moisture content were soaked in water in the fermenter at 20 ℃ and then fine air bubbles were continuously provided by the air pump of the fermenter for 48 h. GBR samples were collected at 4, 8, 12, 16, 24, 32, 40, and 48 h, and non-germinated BR was used as the control (0 h).

### 3.2. Metabolomic Analysis and PLS-DA Score Plots

The GBR metabolite profiles including amino acids, acidic compounds, lipids, and sugars were analyzed using UPLC-Q-TOF MS and GC-MS (Appendix A) and the differences between the GBR samples were visualized by PLS-DA (Figure 3). PLS-DA score plots for the LC-MS and GC-MS data showed that all sample groups were clearly separated from each other with statistically acceptable quality parameters (LC-MS: R_2_X = 0.456, R_2_Y = 0.235, Q_2_ = 0.225, and *p*-value = 9.17 × 10^−10^; GC-MS: R_2_X = 0.963, R_2_Y = 0.238, Q_2_ = 0.230, and *p*-value = 1.73 × 10^−15^). Moreover, the cross-validation values (R_2_ intercept <0.05 and Q_2_ intercept <−0.1 for LC-MS and GC-MS) determined by the permutation test (*n* = 200) indicated that the PLS-DA models were statistically acceptable. In particular, the PLS-DA score plot of the LC-MS-based metabolite analysis showed that the GBR groups in the imbibition stage (0, 4, and 8 h) were clearly separated from the GBR groups at 12–24 h of germination by the second component, t[2], and from the GBR groups toward the end of germination (40 and 48 h) by the first component, t[1]. However, GBR groups in the imbibition stage and at 12–24 h of germination were clustered with each other. Unlike a LC-MS-based PLS-DA score plot, a PLS-DA score plot of the GC-MS-based metabolite analysis showed that the non-germinated sample (control, 0 h) was clearly separated from all GBR samples, whereas the GBR groups at 4–24 and 36–48 h germination times were clustered. Both PLS-DA score plots indicated that GC-MS metabolite profile of the dry BR seed (0 h) was significantly altered by water absorption, but the LC-MS metabolite profiles of the dry BR seeds did not change by imbibition. This result was similar to the principal component analysis score plots of the wheat endosperm metabolite profiles during seed germination (0–24 h) [15] and the polar metabolite profiles including sugars, organic acids, and amino acids during the barley malting process [23].

### 3.3. Identification of Major Metabolites

The *p*-values of all normalized chromatogram intensities of the GBR metabolites obtained by GC-MS and LC-MS analyses were analyzed to identify the metabolites contributing to the observed differences between various groups. Twenty-four metabolites including acidic compounds (lactic acid, phosphoric acid, malic acid, and citric acid), amino acids (alanine, GABA, and glutamic acid), sugars (fructose, glucose, sorbitol, gluconic acid, myo-inositol, sucrose, cellobiose, and maltose), lipid metabolites (glycerol, palmitic acid, and stearic acid), and volatile compounds (perhydrogeraniol, 4,8-dimethylnonanol, 2-isopropyl-5-methylheptanol, 2,4-diethylheptanol, 1,4-diacetylbenzene, and *p*-hydroxydiisopropylbenzene) were identified as the major contributors to the differences in the groups on the PLS-DA score plot for GC-MS data (Table 1). Moreover, eight lipid metabolites [palmitic acid hydrazide, phosphatidylcholine (PC) (16:0/2:0), phytosphingosine, lysophosphatidylcholines (LPCs) (14:0, 16:0, 18:1, and 18:2), and lysophosphatidylethanolamine (LPE) (18:2)] and tryptophan were identified via UPLC Q-TOF MS analysis (Table 2). Variable importance in projection (VIP) values for most metabolites except all flavor compounds, palmitic acid, stearic acid, lactic acid, and alanine were above 0.99, indicating that these metabolites mainly contributed to the separation of the samples on the PLS-DA plot. Among these metabolites, LPCs (14:0, 16:0, and 18:2), sucrose, and glucose were the main metabolites of GBR. The number of GBR metabolites identified using LC-MS and GC-MS was much smaller than those of the metabolites (174 and 173 metabolites for BR kernel and barley, respectively) identified from BR kernel [19] and barley [23] germinated for 96 and 172 h, respectively, which were analyzed using GC-MS or GC-flame ionization detector (FID). However, the metabolite profiles of the BR kernel and barley changed mainly after post-germination, but not until the radicle-emergence stage as observed in the case of GBR.

### 3.4. Relative Abundance of Identified Metabolites and Proposed Metabolomic Pathway

The normalized intensities of all identified metabolites were compared and the metabolomic pathway associated with the germination of BR was proposed Figure 4. It was determined that carbohydrate metabolism, citric acid cycle, and lipid metabolism were the processes that were mainly changed during the germination of BR. Similar results were previously reported for other starch-containing grains such as barley [23] and wheat [24]. In general, large amounts of storage substances including starch granules, lipids, and proteins that are accumulated in the mature seeds are broken down into monosaccharides, fatty acids, and amino acids by many enzymes in the seeds to produce the necessary energy during germination [8].

Among the energy storing components, the carbohydrates varied significantly during germination. The contents of all identified metabolites except fructose and myo-inositol in the dry seed (0 h) decreased considerably by water absorption in the imbibition stage. The amounts of maltose, cellobiose, glucose, sorbitol, and gluconic acid decreased by 11.4, 10, 3, 2.6, 1.7, and 2.3 times, respectively, up to 12 h, compared to those for the non-germinated BR (0 h), and thereafter, increases in their amounts were observed. In particular, the amount of glucose in BR germinated for above 40 h was similar to the initial amounts. Sucrose decreased continuously during the entire germination period and at 48 h, the amount was 7.9 times lower than that of the non-germinated BR, resulting in an accumulation of fructose and myo-inositol. At 48 h, their amounts were 3.9 and 1.5 times higher, respectively, than those of the non-germinated BR. The changes in these carbohydrate profiles for 48 h of germination were lower than those for the long-term germination of BR [19] and barley [23] because α-amylase was mainly produced in the germinating rice seeds after two days of imbibition [25].

In the citric acid cycle–associated metabolism, the amounts of malic acid, GABA, and glutamic acid increased continuously during the entire germination period and at 48 h, their amounts were approximately 60% higher than those of the non-germinated BR, while the amount of citric acid decreased up to 24 h and then increased. Interestingly, the accumulation of GABA, which has various health benefits [26], was much lower than that for the germinated rice under different soaking conditions [5,27]. Many studies have shown that the accumulation of GABA (2.3–24.7 times increase) is significantly dependent on the rice cultivar, germination time, and temperature [5,27].

In the lipid metabolism, the contents of the identified LPCs increased up to 16 h, but a further increase was not observed up to 48 h, whereas LPE(18:2), PC, phytosphingosine, palmitic acid hydrazide, and phosphoric acid increased after the imbibition stage. Glycerol content decreased during germination up to 12 h, and stearic acid and palmitic acid contents increased slightly. Similar to other storage substances, the lipid breakdown was associated with energy production during germination. However, several literature reports have shown that the utilization of lipid stored in the embryonic cells occurs locally during germination and interference with the related pathways negatively affects the germination [28,29].

In addition to the metabolites associated with energy generation, some secondary metabolites including perhydrogeraniol, 4,8-dimethylnonanol, 2-isopropyl-5-methylheptanol, 2,4-diethylheptanol, and 1,4-diacetylbenzene were identified. The amounts of these metabolites did not change significantly during germination; however, in a prior literature report, the phenolic compounds were found to be more abundant in GBR and BR compared to those in white rice [30].

## 4. Conclusions

BR was germinated for up to 48 h to make it suitable for eating. BR absorbed water for up to 8 h (imbibition stage), and BR samples germinated for 8–40 h were in the coleoptile-emergence stage, whereas those germinated for 48 h were in the radicle-emergence stage. To better understand the relationship between germination and nutritional quality of the GBR, the changes in the metabolite profiles during germination were analyzed using LC-MS and GC-MS, and differences in the samples were visualized by PLS-DA. Twenty-five metabolites including acidic compounds, amino acids, sugars, lipid metabolites, and secondary metabolites, which contributed toward the differences in various groups on the PLS-DA score plot were identified. Based on the identified metabolites, a metabolomic pathway was proposed, and it was determined that carbohydrate metabolism, TCA cycle, and lipid metabolism were the main processes that changed during the germination of BR. Of these metabolites, the carbohydrates associated with energy production and lipid metabolites varied significantly. Although further studies are needed on the relationship between metabolite profiles and the nutritional quality of GBR, these results provide useful insights into the effect of germination on the physiological and chemical changes of BR.

## Figures and Tables

**Figure 1 foods-09-01130-f001:**
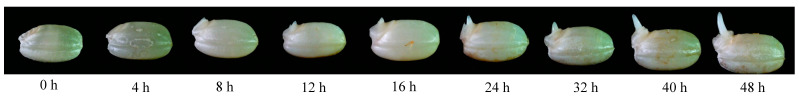
Morphology of germinated brown rice (GBR) according to the germination period.

**Figure 2 foods-09-01130-f002:**
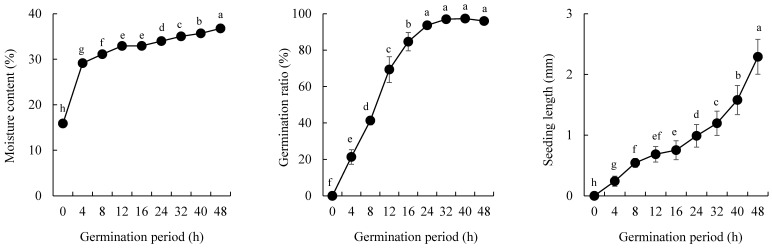
Moisture content, germination ratio, and seedling length of the GBR samples according to the germination period. Different letters on the bars indicate significant differences at *p* < 0.05.

**Figure 3 foods-09-01130-f003:**
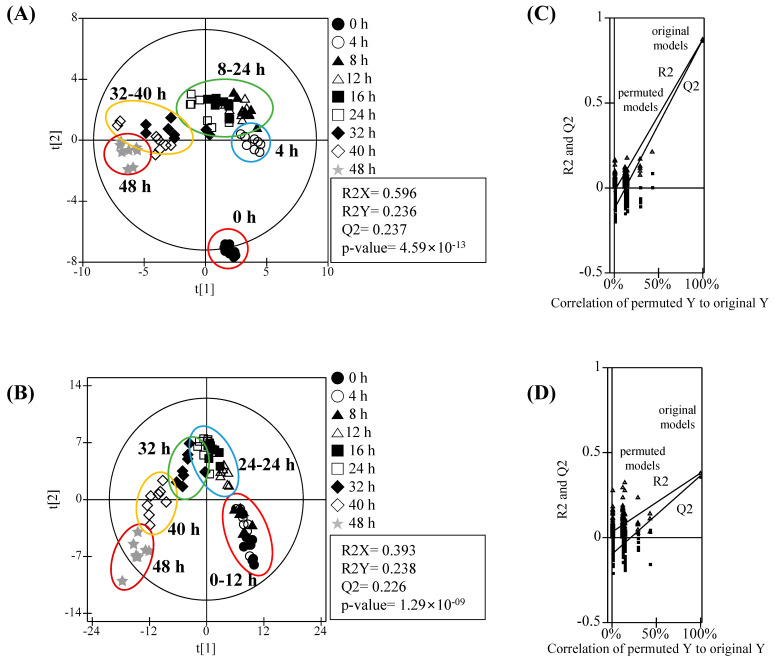
Partial least-squares discriminant analysis (PLS-DA) score plots of GBR metabolites analyzed using (**A**) GC-MS and (**B**) ultra-performance liquid chromatography quadrupole time of flight (UPLC-Q-TOF) MS, and their quality parameters. The qualification of the PLS-DA models was evaluated using R_2_X, R_2_Y, Q_2_, and *p*-values and validated using cross validation with a permutation test (*n* = 200); (**C**) GC-MS and (**D**) LC-MS.

**Figure 4 foods-09-01130-f004:**
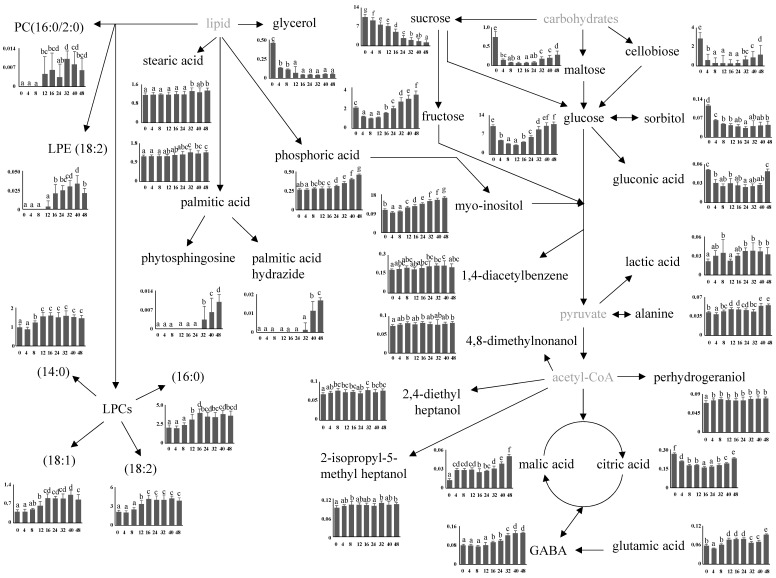
Schematic of the GBR metabolomic pathway and relative abundance of the identified metabolites according to the germination period. Y axis represents the normalized chromatogram intensity, and X axis represents the germination period. Different letters on the bars indicate significant differences at *p* < 0.05.

**Table 1 foods-09-01130-t001:** Identification of GBR metabolites contributing to the separation of the sample groups based on PLS-DA score plot for GC-MS analysis.

RT ^a^ (min)	Compound	RI ^b^	VIP ^c^	*p*-Value ^d^
6.73	4,8-dimethylnonanol	1073	0.46	1.77 × 10^−1^
12.24	1,4-diacetylbenzene	1426	0.5	5.82 × 10^−2^
6.27	lactic acid	1045	0.52	5.98 × 10^−3^
10.26	2-isopropyl-5-methyl-1-heptanol	1292	0.52	8.45 × 10^−3^
21.29	stearic acid	2226	0.56	1.10 × 10^−2^
14.01	*p*-hydroxydiisopropylbenzene	1560	0.61	1.24 × 10^−2^
10.39	2,4-diethyl-1-heptanol	1300	0.62	1.15 × 10^−2^
6.66	perhydrogeraniol	1068	0.67	2.71 × 10^−3^
19.38	palmitic acid	2029	0.69	3.56 × 10^−4^
7.01	alanine	1089	0.88	1.12 × 10^−16^
14.54	glutamic acid	1601	0.90	8.99 × 10^−33^
12.82	malic acid	1469	1.05	8.68 × 10^−30^
17.42/17.54	fructose	1845/1856	1.13	2.23 × 10^−27^
13.42	γ-aminobutyric acid (GABA)	1514	1.14	6.88 × 10^−31^
9.70	phosphoric acid	1256	1.15	2.89 × 10^−38^
19.74	myo-inositol	2065	1.16	8.40 × 10^−35^
18.81	gluconic acid	1973	1.17	1.76 × 10^−22^
24.53	sucrose	2601	1.23	1.18 × 10^−44^
17.70/17.92	glucose	1870/1890	1.29	1.11 × 10^−40^
18.13	sorbitol	1909	1.39	1.43 × 10^−24^
16.86	citric acid	1795	1.40	9.30 × 10^−42^
9.73	glycerol	1258	1.43	1.28 × 10^−38^
25.54	maltose	2731	1.44	1.89 × 10^−29^
25.33	cellobiose	2703	1.45	2.46 × 10^−31^

^a^ RT, retention time; ^b^ RI, retention index; ^c^ VIP, variable importance in the projection; ^d^
*p*-values were analyzed by Duncan’s test.

**Table 2 foods-09-01130-t002:** Identification of GBR metabolites contributing to the separation of the sample groups based on PLS-DA score plot for UPLC-Q-TOF MS analysis.

RT ^a^ (min)	Compound	Exact Mass	MS Fragments	VIP ^b^	*p*-Value ^c^
		(M + H)			
1.65	tryptophan	205.0821	188	0.47	7.32 × 10^−1^
5.37	PC (16:0/2:0)	538.3476	184, 440	0.99	2.15 × 10^−6^
5.79	phytosphingosine	318.2986	282	1.24	1.43 × 10^−12^
6.94	LPC (16:0)	496.3404	184, 104	1.35	2.15 × 10^−13^
6.56	LPE (18:2)	478.2904	337, 460	1.37	4.64 × 10^−19^
7.12	LPC (18:1)	522.3558	184, 104	1.43	5.43 × 10^−17^
6.13	LPC (14:0)	468.3089	184, 104	1.44	8.35 × 10^−14^
5.02	palmitic acid hydrazide	271.2731	254	1.47	3.92 × 10^−28^
6.60	LPC (18:2)	520.3405	184, 104	1.50	1.22 × 10^−17^

^a^ RT, retention time; ^b^ VIP, variable importance in the projection; ^c^
*p*-values were analyzed by Duncan’s test. PC phosphatidylcholine; LPC: lysophosphatidylcholine; LPE: lysophosphatidylethanolamine.

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
