# Peer review of "Metabolomic Analysis of Germinated Brown Rice at Different Germination Stages"

_foods, 2020, doi:10.3390/foods9081130_

Round 1
Reviewer 1 Report
A technically well prepared, thoroughly performed scientific study dedicated to the metabolomic changes of the germinated brown rice. The well done metabolomic study brought minimal surprising metabolic data, owing to the absence of a suitable leading hypothesis corroborating the study. Nevertheless, the results may be of interest to researchers involved in food research, in particular, in the brown rice and its nutrients.
Technical remarks:
1.Why the 50 % MeOH was prefered for the extraction ?
2. All data were normallized using the average mass intensity of the used internal standards. The standards and the method should be described in a greater detail in the text.
Author Response
Response to reviewer 1 comment
Thank you very much for your constructive comments and suggestions.
Point 1: Why the 50% MeOH was preferred for the extraction?
Response: In general, 70-80% methanol is used to extract metabolites from plant based materials, but 80% methanol extracted only most of lipid-related substances from germinated brown rice. Therefore, 50% methanol was used to extract various metabolites from germinated brown rice as well as lipid-related metabolites.
Point 2: All data were normalized using the average mass intensity of the used internal standards. The standards and the method should be described in a greater detail in the text.
Response: As mentioned in the section of ‘Sample preparation for metabolomic analysis’ (page 5 and 6), terfenadine and dicyclohexyl phthalate were used as internal standards for LC/MS and GC/MS, respectively and the methods were described in the section.
Reviewer 2 Report
In this paper Kim et al characterize changes in metabolite profiles in brown rice during germination using a metabolomics approach. In general, the paper seems like it would be a useful addition to the literature. The methods and conclusion are written clearly enough, although the figures need major improvement before publication. A few specific comments:
- A few improvements to English style are needed eg line 43 should be "prevented against hyperlipidemia... etc". The word "among" is used wrongly in some places and should be replaced by a different preposition.
- Table 1: this is simple numeric data and should be replaced with a figure/figure panel with germination period on the x-axis. The superscripts are difficult to read.
- I question the suitability of PLS-DA for this analysis, as there are too many groups and models are not used to predict new observations. The same results could have been obtained in a more simple and interpretable way using PCA and loadings. Time could have been represented as a continuous variable.
- In Figure 2, (b), (c) and (d) are cut off the page and I was unable to read them.
- Table 2 would be more informative ordered by VIP or p-value rather than chemical properties.
- Table 3 has problems with formatting (tryptophan)
- Figure 3 is too small and mostly unreadable.
Author Response
Response to reviewer 2 comment
Thank you very much for your constructive comments and suggestions.
Point 1: A few improvements to English style are needed eg line 43 should be “prevented against hyperlipidemia…etc”. The word “among” is used wrongly in some places should be replaced by a different preposition.
Response 1: The manuscript has been carefully revised by a professional language editing service to improve the grammar and readability. The sentence of line 43-46 has been modified as ‘the intake of GBR prevented against hyperlipidemia, hypertension, and hyperglycemia, and showed a positive correlation to the reduction in the risk of some chronic diseases such as cancer, diabetes, cardiovascular disease, and Alzheimer's disease’.
Among (line 3, 75, 157, 215, 280, 282, table 1, and table 2) à in
Among (line 221 and 285) à of
Point 2: Table 1, this is simple numeric data and should be replaced with a figure/figure panel wit germination period on the x-axis. The superscripts are difficult to read.
Response 2: Table 1 is shown as a figure. The numbers for Figures and Figure caption have been rearranged. Also, the sentence ‘The general characteristics of the GBR samples in terms of germination ratio, seedling length, and moisture content are summarized in Table 1 and their morphologies are shown in Figure 1.’(page 8, line 167-168) has been changed to ‘The morphologies of the GBR samples are shown in Figure 1 and their general characteristics in terms of germination ratio, seedling length, and moisture content are shown in Figure 2.’.
Figure 2. Moisture content, germination ratio, and seedling length of the GBR samples according to the germination period. Different letters on the bars indicate significant differences at p < 0.05. |
Point 3: I question the suitability of PLS-DA for this analysis, as there are too many groups and models are not used to predict new observations. The same results could have been obtained in a more simple and interpretable way using PCA and loadings. Time could have been represented as a continuous variable.
Response 3: The results of PCA score plots is similar with those of PLS-DA score plots, but PLS-DA score plots more clearly showed the discrimination according to time compared to PCA. To clearly show the result, the PLS-DA score plots were modified.
PCA score plots for GBR samples analyzed using GC/MS (A) and LC/MS (B) |
Modified PLS-DA score plots for GBR samples analyzed using LC/MS and GC/MS |
Point 4: In Figure 2, (b), (c) and (d) are cut off the page and I was unable to read them.
Response 4: In the process of converting to PDF file, the quality of the figure seems to have deteriorated. An original figure with high quality will be provided as a zip file.
Point 5: Table 2 would be more informative ordered by VIP or p-value rather than chemical properties.
Response 5: Table 2 and 3 have been rearranged by VIP values.
Point 6: Table 3 has problems with formatting (tryptophan).
Response 6: There is no problem with the format. The abbreviation is capitalized and other material names, such as tryptophan, are written in lowercase.
Point 7: Figure 3 is too small and mostly unreadable.
Response 7: An original figure with high quality will be provided as a zip file.